# Overlaid positive and negative feedback loops shape dynamical properties of PhoPQ two-component system

**Satyajit D Rao**[1], **Oleg A Igoshin**[1,2]*

**1** Department of Bioengineering, Rice University, Houston, Texas, USA, **2** Departments of Biosciences and Chemistry, Center for Theoretical Biological Physics, Rice University, Houston, Texas, USA

* igoshin@rice.edu

**Data Availability Statement:** All relevant data are within the manuscript and its Supporting information files, and all codes and parameter/data sets can be found in the following GitHub

## Abstract

Bacteria use two-component systems (TCSs) to sense environmental conditions and change gene expression in response to those conditions. To amplify cellular responses, many bacterial TCSs are under positive feedback control, i.e. increase their expression when activated. *Escherichia coli* $Mg^{2+}$-sensing TCS, PhoPQ, in addition to the positive feedback, includes a negative feedback loop via the upregulation of the MgrB protein that inhibits PhoQ. How the interplay of these feedback loops shapes steady-state and dynamical responses of PhoPQ TCS to change in $Mg^{2+}$ remains poorly understood. In particular, how the presence of MgrB feedback affects the robustness of PhoPQ response to overexpression of TCS is unclear. It is also unclear why the steady-state response to decreasing $Mg^{2+}$ is biphasic, i.e. plateaus over a range of $Mg^{2+}$ concentrations, and then increases again at growth-limiting $Mg^{2+}$. In this study, we use mathematical modeling to identify potential mechanisms behind these experimentally observed dynamical properties. The results make experimentally testable predictions for the regime with response robustness and propose a novel explanation of biphasic response constraining the mechanisms for modulation of PhoQ activity by $Mg^{2+}$ and MgrB. Finally, we show how the interplay of positive and negative feedback loops affects the network's steady-state sensitivity and response dynamics. In the absence of MgrB feedback, the model predicts oscillations thereby suggesting a general mechanism of oscillatory or pulsatile dynamics in autoregulated TCSs. These results improve the understanding of TCS signaling and other networks with overlaid positive and negative feedback.

## Author summary

Feedback loops are commonly observed in bacterial gene-regulatory networks to enable proper dynamical responses to stimuli. Positive feedback loops often amplify the response to stimulus, whereas negative feedback loops are known to speed-up the response and increase robustness. Here we demonstrate how combination of positive and negative feedback in network sensing extracellular ion concentrations affects its steady-state and

repository: https://github.com/satyajitdrao/
PhoPQManuscript.git.

**Funding:** The research was supported by Welch
Foundation Grant C-1995, National Science
Foundation grant MCB-1616755 to OAI and is NSF
PHY 201974 award for the Center of Theoretical
Biological Physics. The funders had no role in
study design, data collection and analysis, decision
to publish, or preparation of the manuscript.

**Competing interests:** The authors have declared
that no competing interests exist.

dynamic responses. We utilize published experimental data to calibrate mathematical
models of the gene regulatory network. The resulting model quantitatively matches exper-
imentally observed behavior and can make predictions on the mechanism of negative
feedback control. Our results show the advantages of such a combination of feedback
loops. We also predict the effect of their perturbation on the steady-state and dynamic
responses. This study improves our understanding of how feedback loops shape dynam-
ical properties of signaling networks.

## Introduction

Bacteria use two component systems (TCSs) to sense and respond to environmental stimuli [1,
2]. TCSs are also widely used in synthetic biology applications to sense specific stimuli and
control gene expression [3–5]. A TCS consists of a sensor kinase often located on the inner
membrane and a cognate response regulator protein located in the cytoplasm. The sensor
kinase senses environmental stimulus and responds by autophosphorylating at the histidine
residue [6]. Phosphorylated kinase catalyzes a transfer of phosphate to the response regulator.
In the absence of activating conditions, sensor kinases sometimes have phosphatase activity,
i.e. they can dephosphorylate the response regulator. The phosphorylated response regulator is
transcriptionally active, initiating cellular response. As part of cellular response, the response
regulator often activates transcription of genes encoding the two components themselves [7],
creating a positive feedback loop.

The $Mg^{2+}$ -sensing PhoPQ TCS is found in many bacterial species, such as *Salmonella*, *Yer-
sinia pestis* and *E. coli* [8–14]. The sensor kinase PhoQ responds to low extracytoplasmic $Mg^{2+}$
levels, acidic pH and antimicrobial peptides. In high $Mg^{2+}$, the periplasmic sensing domain of
PhoQ is bound to $Mg^{2+}$ resulting in a conformation of PhoQ that has low autokinase activity
but high phosphatase activity towards phosphorylated PhoP (PhoP-P) [8]. That keeps the
expression of PhoP-P-dependent genes low. In response to $Mg^{2+}$ limitation, dissociation of
$Mg^{2+}$ from PhoQ promotes a conformational change that increases the autokinase activity and
suppresses the phosphatase activity [15]. That leads to accumulation of PhoP-P and increase in
the expression of its regulon. PhoP-P regulons vary significantly between different bacterial
species but retain a few common features. First, the PhoPQ TCS upregulates transcription of
its own operon *phoPQ*. This upregulation leads to a positive feedback in the system. Second,
PhoP activates transcription of a small integral membrane protein, MgrB in *E. coli*, *Klebsiella
pneumoniae*, *Salmonella*, *Yersinia pestis* that limits kinase activity (Fig 1) [16–20]. These inter-
actions form a negative feedback loop.

How does positive autoregulation of PhoP/PhoQ affect phosphorylation level of PhoP?
Notably, over a range of low $Mg^{2+}$ concentrations, elimination of autoregulation of PhoPQ
results in no significant difference in the PhoP-P activity measured via transcriptional reporter
of the *mgrB* promoter [21]. This observation suggests that PhoP-P level is insensitive (robust)
to increase of *phoPQ* operon production. This robustness was confirmed by monitoring PhoP
activity reporter in a strain with a chemically inducible *phoPQ* promoter (Fig 1); increasing
PhoPQ expression post wild-type level does not change the reporter level [21]. Previously pub-
lished mathematical models show that this robustness arises due to bifunctionality of the
kinase [21–23]. However, these models may not be directly applicable for *E. coli* PhoPQ TCS
as they do not account for PhoPQ interactions with MgrB protein. While the exact mechanism
by which MgrB modulates PhoQ activity is unknown, MgrB specifically inhibits kinase activity
through direct interaction [17]. A strain lacking both the *mgrB* gene and PhoQ-phosphatase

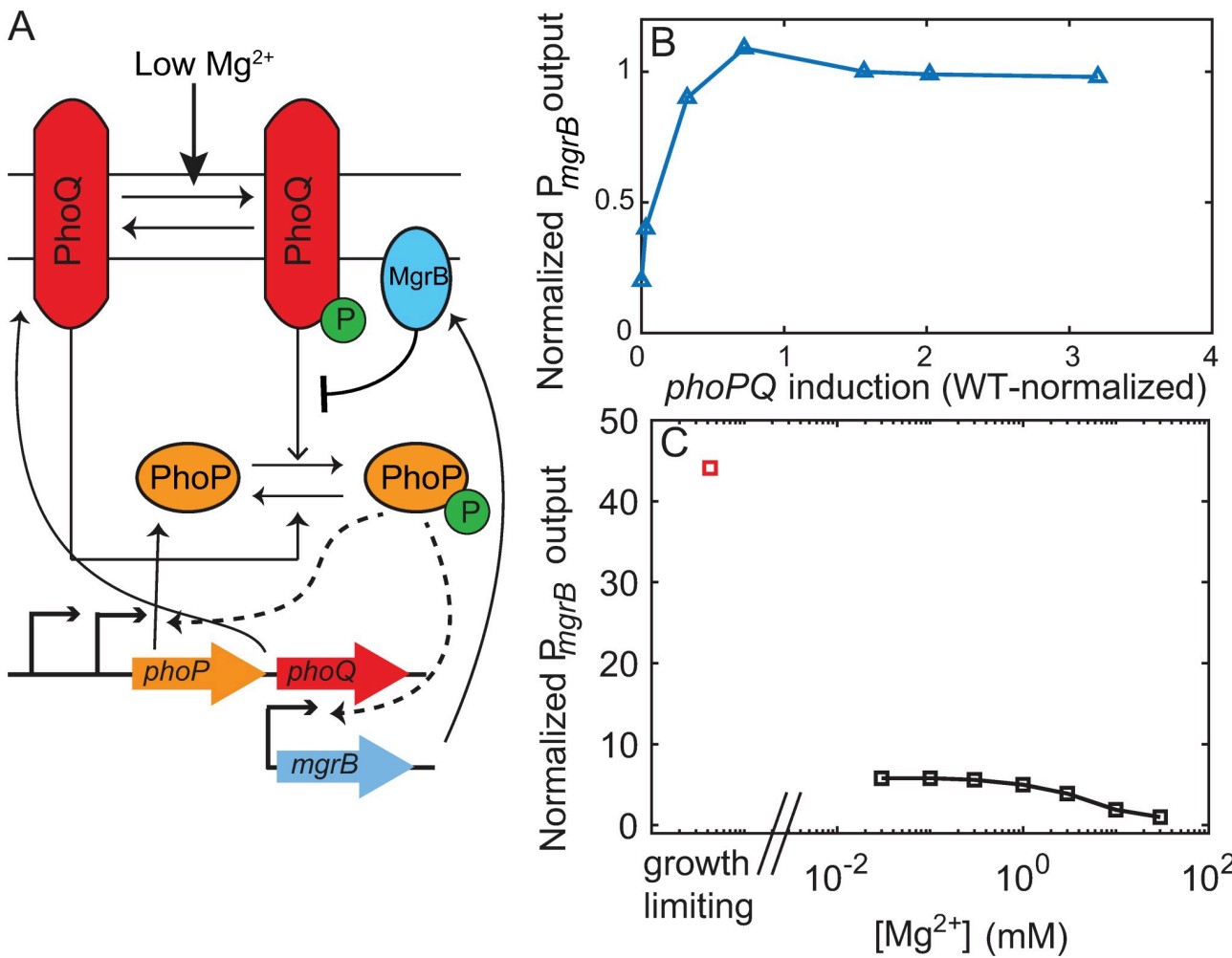

**Fig 1. PhoPQ two component system senses low Mg$^{2+}$ concentrations through direct interactions with PhoQ.** (A) When activated by low external Mg$^{2+}$ PhoQ undergoes autophosphorylation, transfers phosphoryl group to PhoP which activates transcription of downstream genes. PhoP positively regulates transcription of *phoPQ* operon, as well as mgrB. MgrB binds PhoQ and suppresses kinase activity. (B) Normalized reporter output from $P_{mgrB}$ saturates as *phoPQ* operon transcription is increased. (C) Steady state normalized reporter output ($P_{mgrB}$) plateaus as Mg$^{2+}$ decreases, but increases further at growth limiting conditions (hypothetical normalized reporter output at growth limiting Mg$^{2+}$, red square). Plot recreated from [21].

activity shows higher promoter activity compared to a strain merely lacking the PhoQ-phosphatase activity [16]. Since overexpressing PhoQ could in-principle outcompete the inhibitory effect of MgrB, understanding robustness of PhoP-P to PhoPQ overexpression requires models that explicitly include negative feedback regulation.

Notably, robustness of PhoP activity to elimination of PhoPQ autoregualtion is no longer observed in growth-limiting Mg$^{2+}$ levels ($<10^{-3}$ mM) [21]. Furthermore, in these conditions, the PhoP activity greatly exceeds the activity observed over a range of low but not growth-limiting (between 1 and 0.01 mM) Mg$^{2+}$ concentrations. Notably: PhoP activity is nearly the same over that range of Mg$^{2+}$ levels forming a plateau between 1 and 0.01 mM Mg$^{2+}$ following a gradual increase from 100 to 1 mM Mg$^{2+}$. Such a plateau has been observed for multiple promoters with varying affinities to PhoP-P, suggesting this pattern is not a property of one particular promoter [9]. Interestingly, at 0.01 mM Mg$^{2+}$ the levels of PhoP-P are such that the promoters remain far from saturation [9].

A combination of plateau in promoter activity over a range of low $Mg^{2+}$ levels with further increased activity in growth-limiting $Mg^{2+}$ can be referred to as biphasic dose-response (Fig 1). Miyashiro and Goulian hypothesize that this biphasic dose-response is indicative of $Mg^{2+}$ binding to PhoQ at multiple sites with different affinities [9]. However, this hypothesis has not been tested experimentally or theoretically. Alternatively, feedback architecture might shape a biphasic dose-response. If negative feedback dominates over a range of low $Mg^{2+}$ concentrations, while positive autoregulation is strongly activated only in growth-limiting $Mg^{2+}$, we could perhaps expect steady state PhoP-P to display two phases of activation. Since it is unclear how overlaid positive and negative feedback loops shape observed dose-response, detailed mathematical models of PhoPQ TCS can be used to understand steady state PhoP-P as a function of $Mg^{2+}$.

In this study, we use mathematical modeling to understand how positive and negative feedback loops interact to shape dynamical properties of the PhoPQ TCS in *E. coli*. First, we identify conditions under which PhoP-P remains robust to *phoPQ* overexpression even in presence of MgrB-mediated negative feedback. Next, we search for mechanisms underlying the biphasic dose-response. We use published temporal and steady state data for wild type and mutant *E. coli* strains to calibrate our models. Finally, we use these calibrated models to understand advantages of the overlaid positive and negative feedback design of the PhoPQ system. Taken together, this study shows how mathematical modeling and experimental data can be used together to understand the relationship between network structure and cellular function in bacteria.

## Results

### PhoPQ TCS can show robustness if MgrB is in excess of PhoQ

Structural sources of robustness to variation in species concentrations have been identified previously for mass-action reaction networks [23, 24]. Two component systems (TCSs) with bifunctional kinases are known examples of such biochemical networks. The concentration of phosphorylated response regulator can be robust to changes in total concentrations of sensory kinase and response regulator proteins [22–24]. To ascertain if the biochemical reaction network of PhoPQ TCS meets the criteria for absolute concentration robustness (ACR) put forth by Shinar and Feinberg [24], we analyze the reaction network with or without MgrB (S1 Text). While a reaction network without MgrB did in fact meet the criteria to obtain ACR in the limit of negligible auto-dephosphorylation, the reaction network of PhoPQ with MgrB network did not (S1 Text). This analysis suggests that in contrast to a typical TCS featuring a bifunctional kinase, robustness to total protein concentrations is not theoretically predicted by the structure of the reaction network. To understand that result we note that ACR occurs due to ability of PhoQ to control both phosphorylation and dephosphorylation flux to PhoP. Increase in PhoQ level will increase both fluxes proportionally without affecting PhoP-P. On the other hand, when MgrB-mediated inhibition of kinase activity is present, increase in PhoQ concentration disproportionally increases dephosphorylation flux.

To find conditions under which PhoP-P could be robust to total-PhoPQ expression, we modify the PhoPQ model in ref [21] to explicitly include negative feedback regulation. We consider a system without positive feedback, i.e. with total-PhoPQ expression controlled independent of PhoP-P. To simplify steady state analysis of PhoP-P, we follow the approach used in ref [21] and break the model into two modules (Fig 2A). For the transcription module the input is PhoP-P and the output is total-MgrB. We use a standard Hill-function to describe how transcription rate of *mgrB* and correspondingly total MgrB concentration depends

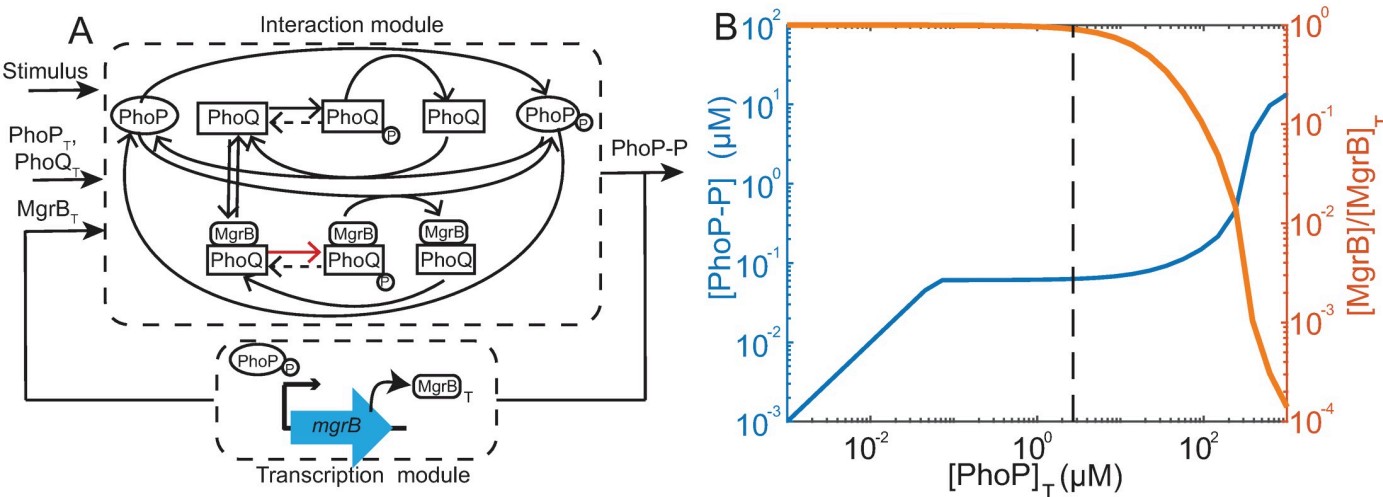

**Fig 2. PhoP-P is robust to overexpression of PhoP, PhoQ.** A—Modified PhoPQ TCS interaction network shows PhoQ binding MgrB, repressing PhoQ autophosphorylation (red arrow marks suppressed rate compared to unbound PhoQ autophosphorylation). This forms the interaction module. The interaction module takes 3 inputs (i) Stimulus (autophosphorylation rate), (ii) Total PhoP (PhoQ is assumed proportional, and 1/40 times PhoP based on actual measurements) and (iii) Total MgrB. Total protein is represented by subscript T in all figures and text. The system is numerically solved for a steady state concentration of PhoP-P as a function of varying PhoP, PhoQ total. The interaction module is coupled with a transcription module representing negative feedback with PhoP-P as input and total MgrB concentration as output. B—The system is solved numerically for steady state concentration of PhoP-P as a function of varying PhoP (and PhoQ) total (blue). PhoP-P is robust to PhoP/PhoQ concentrations, increasing further when PhoQ concentration is large enough to overcome MgrB negative feedback. Over most of the range of PhoQ concentrations, MgrB ≈ MgrB-total indicating large stoichiometric excess MgrB (orange line). Robustness breaks when MgrB is no longer in large excess of PhoQ. Dashed line indicates PhoP concentration estimated from measurements of Ref. [25].

[PhoP-P](Eq 1).

$$[\mathrm{MgrB}]_T = [\mathrm{MgrB}]_T^0 \left( 1 + \frac{f_B[\mathrm{PhoP\text{-}P}]^2}{K_B^2 + [\mathrm{PhoP\text{-}P}]^2} \right) \tag{1}$$

Here $[\mathrm{MgrB}]_T^0$ denotes minimum MgrB concentration at basal expression of *mgrB*, $f_B$ represents maximal fold upregulation of MgrB while $K_B$ denotes half activation concentration of PhoP-P. The interaction module consists of the cycle of phosphorylation-dephosphorylation catalyzed by PhoQ and PhoQ-MgrB. For the interaction module the inputs are total-PhoP, PhoQ and MgrB as well as the stimulus level (i.e. autophosphorylation rate of PhoQ). Steady state is at the intersection of the two modules (S1 Fig). As in the model in ref [21], we assume total-PhoP/PhoQ ratio is constant, and this ratio is greater than 1 [25]. To model MgrB inhibition, we assume that when MgrB binds to PhoQ, its autophosphorylation rate (Fig 2A) decreases by a factor $\lambda \leq 1$ and phosphatase activity increases by a factor $\gamma \geq 1$ to allow for the possibility that MgrB might enhance phosphatase activity of PhoQ. The remaining rate constant parameters are assumed same for PhoQ and PhoQ-MgrB states.

With this model we investigate how steady state [PhoP-P] depends on total PhoP, PhoQ (see S2 Text for full analysis). Basal expression of MgrB ($[\mathrm{MgrB}]_T^0$ in Eq 1) is a free parameter. For each value of $[\mathrm{MgrB}]_T^0$ we can solve the two modules for a range of total-PhoP, PhoQ values. We find that steady state [PhoP-P] is not in general robust (S1 Fig). However, at high $[\mathrm{MgrB}]_T^0$ values, PhoP-P can be robust over a limited range of total-PhoP (Fig 2B).

To understand how steady state PhoP-P can show robustness to only a limited range of total-PhoP, and only at high $[\mathrm{MgrB}]_T^0$ values, we look at phosphorylation and dephosphorylation

fluxes of PhoP (S2 Text). Equating phosphorylation and dephosphorylation fluxes, we can obtain an expression for [PhoP-P] shown in Eq 2

$$[\text{PhoP-P}] \approx C_p \frac{\left(1 + \lambda \frac{[\text{MgrB}]}{K_D}\right)}{\left(1 + \gamma \frac{[\text{MgrB}]}{K_D}\right)} \tag{2}$$

In this equation, [PhoP-P] depends on concentration of free MgrB. Since concentration of free MgrB generally depends on concentration of PhoQ, there is no robustness. However, when MgrB is in large excess of PhoQ, the function simplifies to Eq 3.

$$[\text{PhoP-P}] \approx C_p \frac{\left(1 + \lambda \frac{[\text{MgrB}]_T}{K_D}\right)}{\left(1 + \gamma \frac{[\text{MgrB}]_T}{K_D}\right)} \tag{3}$$

Here $C_p$ is a combination of parameters as noted previously in refs [21, 22], and $K_D$ is the dissociation constant for [PhoQ-MgrB]. In this expression, [PhoP-P] then depends only on MgrB-total, which in turn depends on [PhoP-P] (Eq 1). Thus, steady state [PhoP-P] remains independent of total PhoQ, PhoP. Indeed, output ceases to be robust once $[\text{PhoQ}]_T \sim [\text{MgrB}]_T$. This is illustrated by a plot of [MgrB]/[MgrB]$_T$ (Fig 2B). Notably, the robustness to PhoP/PhoQ overexpression can be observed despite variations in $\lambda$, $K_D$ or the ratio of PhoP: PhoQ (S2 Fig). Thus our model of PhoPQ-MgrB with negative feedback shows that robustness of [PhoP-P] to changes in total concentrations of PhoP/PhoQ is not due to the cycle of phosphorylation alone, but can be obtained if MgrB is much more abundant than the kinase PhoQ.

## Models with autophosphorylation suppression by MgrB alone cannot explain biphasic dose-response

Given that the previous model of autophosphorylation suppression by MgrB can explain robustness of PhoP-P to total-PhoPQ levels, we explore steady state dose-response behavior of the model. Specifically, we investigate whether the model can recreate a biphasic dose-response, i.e. show an intermediate plateau (Fig 1C). To compare with experimental data of ref [21], we construct a detailed dynamic model with two reporter proteins YFP and CFP [9] (S5 Text). To calibrate the model, we fit simulated values of YFP:CFP to reported values from various experiments (Methods). Time-course measurements in wild-type cells switched from high to low Mg$^{2+}$ levels (published in [16]) were used to fine tune temporal parameters. The Mg$^{2+}$ step down experiment was also conducted with several mutant strains. Measurements from these mutant strains can serve as important biological constraints on the model. Thus, we simulate YFP:CFP values with *in-silico* mutants and fit to respective experimental values (Methods, S3 Fig).

Steady state values of YFP:CFP over a range of Mg$^{2+}$ concentrations have also been measured for wild-type cells (published in [21]). We use these measurements to tune steady state parameters. In addition, we introduce a qualitative condition for greater reporter output at very high stimulus to recapitulate effects at growth limiting Mg$^{2+}$ [21]. Normalized experimental data used, simulation protocols and parameter fitting procedure are described in Methods.

Over multiple parameter fitting attempts, the model could only show graded increase followed by a plateau in steady state promoter activity (S4 Fig). Thus, we hypothesize that models of PhoPQ-MgrB with more complex mechanisms are required to explain biphasic dose-response.

### Models representing PhoQ with separate kinase and phosphatase conformations can explain biphasic dose-response

To explain biphasic dose-response, we constructed a model of PhoQ with an explicit $Mg^{2+}$ sensing mechanism (S5 Text). While understanding of how $Mg^{2+}$ modulates PhoQ activity is still incomplete in *E. coli*, research in *Salmonella* has suggested that a conformation change resulting from $Mg^{2+}$ binding to PhoQ increases phosphatase activity [8]. Based on this finding we hypothesize two conformations of PhoQ: phosphatase (PhoQ) and kinase (PhoQ*) (Fig 3A). Extracellular $Mg^{2+}$ binds to PhoQ* and drives a transition to PhoQ thus shutting off PhoP-P activity. We assume that extracellular $Mg^{2+}$ concentration does not change over time and include it in the rate constant of switching from PhoQ* to PhoQ (Fig 3A) i.e. $k_{-1} = k^0_{-1}[Mg^{2+}]$. For simplicity we assume that only the kinase conformation undergoes autophosphorylation and phosphotransfer steps, while only phosphatase conformation dephosphorylates PhoP-P. MgrB can bind both conformations of PhoQ independently, subsequently modulating one or more rates.

The mechanism for MgrB mediated suppression of PhoQ remains unknown. Thus, we sought to understand which combination of rates of the phosphorylation cycle is likely to be modulated by MgrB. To this end, we implemented multiple models with MgrB affecting different rate constants in each. With each model we simulate a time course of YFP:CFP following downshifts in $Mg^{2+}$ for all the pairs of downshifts reported in Salazar et al [16] (S3 Fig). Time course simulations are performed for wild-type and *in-silico* mutants (S5 Text). In addition, we perform steady state dose-response simulations. We then obtain parameters that generate close fit with experimental data (Methods). We verify the accuracy of these parameters by performing $Mg^{2+}$ downshift simulations with an *in-silico* mutant expressing *mgrB* constitutively, as well as a PhoQ-phosphatase activity lacking mutant (S5 Fig). If simulations qualitatively

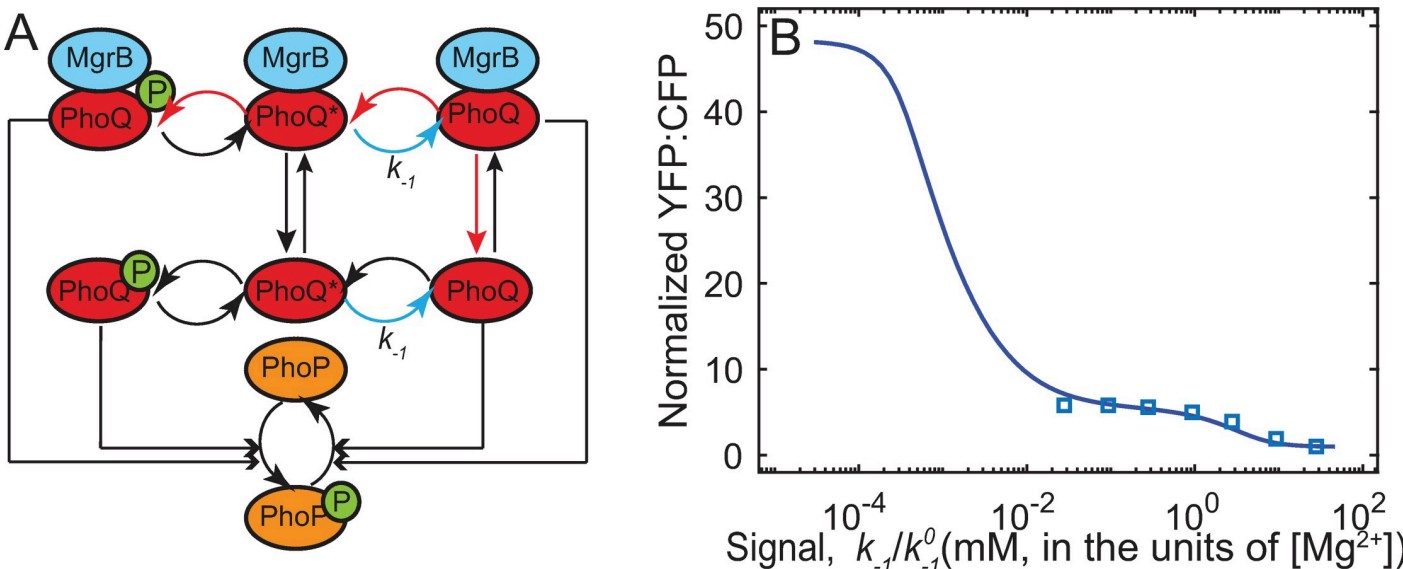

**Fig 3. A two-state model of PhoPQ TCS can explain biphasic response.** A -Schematic of the two-state model. PhoQ exists in phosphatase (PhoQ) or kinase (PhoQ*) form, PhoQ* assumed to bind $Mg^{2+}$ and switch to PhoQ. Concentration of $Mg^{2+}$ in medium assumed constant, and absorbed into a pseudo-first order kinetic rate, $k_{-1}$ (blue arrow). MgrB reversibly binds PhoQ/PhoQ*. B—Simulated output (normalized YFP:CFP; Methods) from the ODE model representing schematic in A with two rate constants suppressed in MgrB bound PhoQ. The pre-factor $k^0_{-1}(s^{-1}mM^{-1})$ converts $Mg^{2+}$ concentration (mM) to rate constant $k_{-1}(s^{-1})$. The affected rates are denoted by red arrows: switching rate from phosphatase to kinase (i.e. $k_1$, PhoQ-$Mg^{2+}$ dissociation), and autophosphorylation. Detailed balance condition is satisfied by assuming PhoQ-MgrB dissociation is suppressed by the same factor as $k_1$. Simulated steady state output shows biphasic response to increasing signal.

match experimental data for the two mutant strains, we consider those parameters for further analysis. In dose-response simulations, we look for a second phase of strong promoter activation at very low $Mg^{2+}$ ($10^{-4}$ mM).

We find the biphasic dose-response pattern (Fig 3B) and closest matches with all experimental data (S6 Fig) in models where MgrB suppresses two rates—(i) autophosphorylation and (ii) activation (PhoQ $\rightarrow$ PhoQ$^*$ transition; red arrows, Fig 3A). Simulated steady state reporter output as a function of signal ($k_{-1}$) shows two distinct ranges of signal where output increases, separated by a plateau (Fig 3B). Notably, the specific value of the output at growth-limiting $Mg^{2+}$ relative to the plateau level does not seem to affect the model's ability to explain biphasic dose response (S7 Fig). Furthermore, this model is able to fit time-course data for wild-type and mutant strains as well (S6 Fig). Interestingly, models with any other combinations of rate constants modulated by MgrB are unable to reproduce this biphasic response to signal. Thus, our analysis isolates a potential mechanism for MgrB suppressing PhoQ kinase activity. Taken together, models with an explicit $Mg^{2+}$ sensing mechanism and where MgrB modulates PhoQ phosphatase to kinase transition and autophosphorylation rates can explain the biphasic dose-response.

Additionally, to check the validity of our conclusions regarding the robustness to PhoP/ PhoQ overexpression we repeated the analysis for two-state model. We find that the model shows a range in which variations in PhoP/PhoQ concentrations do not lead to significant changes in PhoP-P (S8 Fig). Furthermore, the model is able to fit the experimentally measured data on the response of $P_{mgrB}$ promoter to overexpression of *phoPQ* operon under inducer control reported by Miyashiro and Goulian [21].

## Abundance of MgrB, strong suppression and slow transitions between PhoQ states together can create plateau in signal response

While this model can explain biphasic signal response, the mechanism behind a plateau at intermediate signal levels is not fully clear. To understand how steady state [PhoP-P] is insensitive to signal ($k_{-1}$) in our model (Fig 4D), we simplify the model so that analytical solutions will be possible in different ranges of signal. We note that the biphasic response is not generic outcome of the structure of the model, but arises in specific parameter ranges. Our goal is to use analytical approximations to identify the parameter regimes in which biphasic dose-response is possible. Matching phosphorylation and dephosphorylation fluxes can then provide expressions for steady state PhoP-P (see S4 Text for complete analysis). Analyzing how these fluxes change as a function of signal ((Fig 4B and 4C) can clarify the mechanism behind PhoP-P plateau at $k_{-1}$ values corresponding to 1-0.01 mM $Mg^{2+}$ range (Fig 4D).

Phosphorylation and dephosphorylation fluxes depend on the concentrations of the two catalytic states of PhoQ—kinase and phosphatase—and their MgrB-bound counterparts (Fig 4A). In our model we find that over high to intermediate $Mg^{2+}$, nearly all PhoQ molecules exist in MgrB-bound phosphatase state (PhoQ.MgrB$_{ph}$, Fig 4E). We analyze steady state concentrations of PhoQ forms at the limit of complete suppression (S4 Text). We consider the limit in which MgrB binds PhoQ and completely suppresses autophosphorylation (Fig 4A) and activation rate constant (gray arrows Fig 4A). PhoQ.MgrB$_{ph}$ concentration only reduces by dilution due to growth. In our model we find that over high to intermediate $Mg^{2+}$, nearly all PhoQ molecules exist in MgrB-bound phosphatase state (PhoQ.MgrB$_{ph}$, Fig 4E). Therefore, we neglect contribution of PhoQ$_{ph}$ to dephosphorylation of PhoP-P. Further, we assume that all the phosphate that enters the systems through autophosphorylation of PhoQ$^*$ transfers to PhoP (i.e. autodephosphorylation of PhoQ-P is negligible). The net phosphorylation flux of PhoP, then, is equal to autophosphorylation flux (proportional to PhoQ$^*$). Dephosphorylation

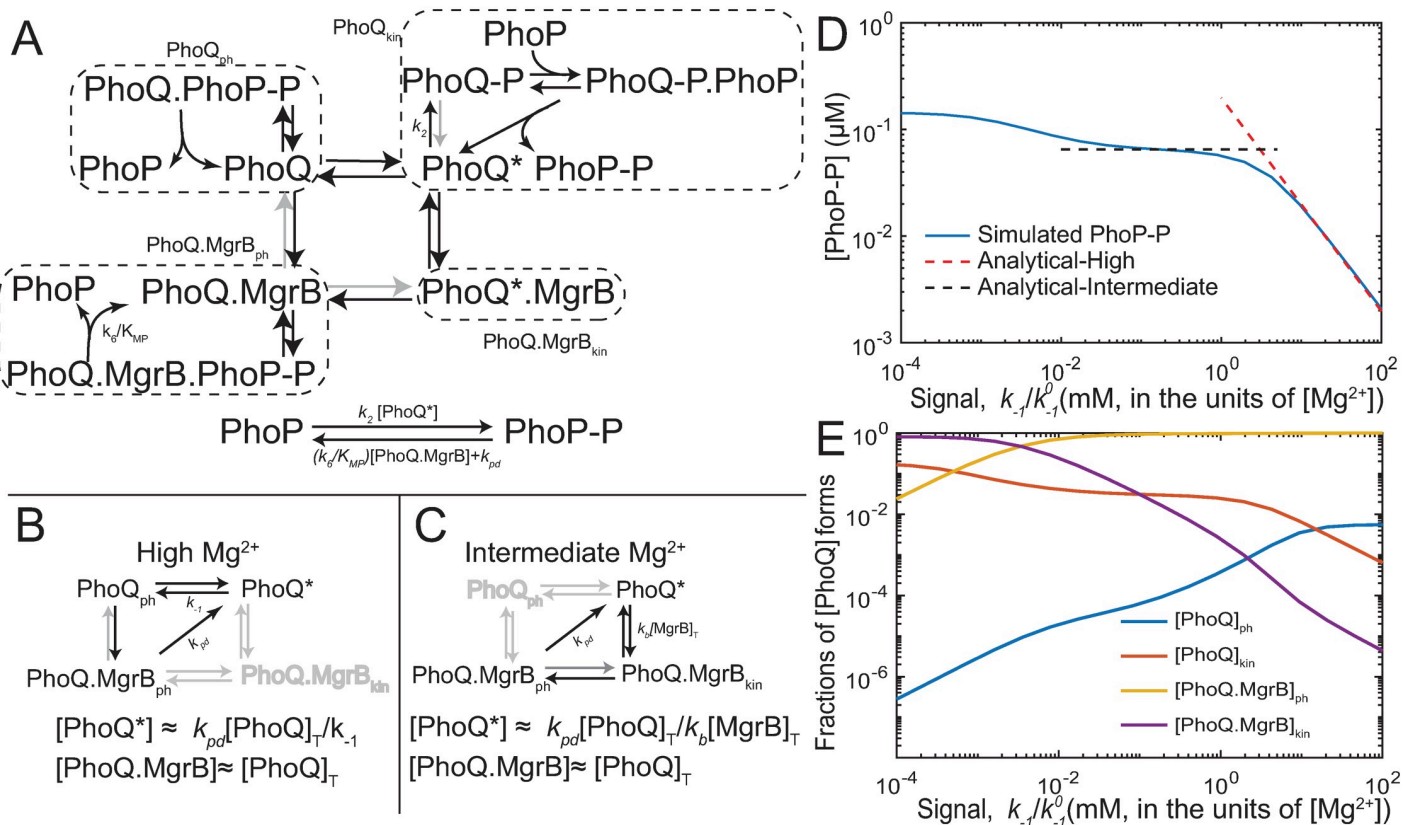

**Fig 4. Understanding the biphasic dose-response possible in the two-state model of PhoPQ.** A—Reactions in PhoPQ-MgrB network. Dotted squares enclose 4 sub forms of PhoQ ($Q_{ph}$, $Q_{kin}$, $QB_{ph}$, $QB_{kin}$). 4 reactions outside the dotted squares have rates comparable to dilution due to growth. Each sub form dilutes with a rate $k_{pd}$, while synthesis is only in the $Q^*$ form. B—Most significant fluxes at high $Mg^{2+}$. Steady state [PhoP-P] is approximated by matching phosphorylation and dephosphorylation fluxes. Phosphorylation flux is proportional to $1/k_{-1}$, while dephosphorylation flux is approximately constant. C- Most significant fluxes at intermediate $Mg^{2+}$. Phosphorylation flux is proportional to $[B]_{Total}$ and independent of $k_{-1}$ while dephosphorylation flux is still approximately constant D- [PhoP-P] as a function of signal showing plateau at intermediate $Mg^{2+}$. Blue line indicates simulated [PhoP-P] from the model in the previous section. Red dashed line shows approximate [PhoP-P] in the high $Mg^{2+}$ range, and black dashed line shows the approximate plateau value of [PhoP-P] in the intermediate $Mg^{2+}$ range. E- Fractions of total Q in the 4 catalytic forms.

flux has two portions—phosphatase activity (proportional to [PhoQ.MgrB]), and dilution due to growth. The steady state [PhoP-P] is found by equating phosphorylation and dephosphorylation fluxes (S4 Text).

Thus, PhoP-P concentration depends on how [PhoQ*] and [PhoQ.MgrB] change as a function of $k_{-1}$. This in turn depends on which fluxes dominate in signal ranges corresponding to high and intermediate $Mg^{2+}$. We find that at high $Mg^{2+}$, flux of [PhoQ*] deactivation (rate $k_{-1}$) dominates over binding MgrB (rate $k_b$[MgrB]; Fig 4B). In this range [PhoP-P] depends linearly on signal. An approximate analytical expression over a range of $k_{-1}$ values corresponding to high $Mg^{2+}$ matches simulated [PhoP-P] using parameters from the previous section (Fig 4D). Thus from high to intermediate $Mg^{2+}$, the promoter output increases a few fold (Fig 3B). At intermediate $Mg^{2+}$, however, flux of PhoQ* binding MgrB is much greater than deactivation (Fig 4C). In this range, [PhoP-P] depends on $[MgrB]_T$ (which in turn depends on [PhoP-P]) but not signal, Fig 4C. An analytical solution for a single steady state [PhoP- P]$_{int}$ independent of signal can be found (Fig 4D). Taken together, strong suppression of kinetic rates by excess MgrB and growth dilution shape biphasic dose response of PhoPQ.

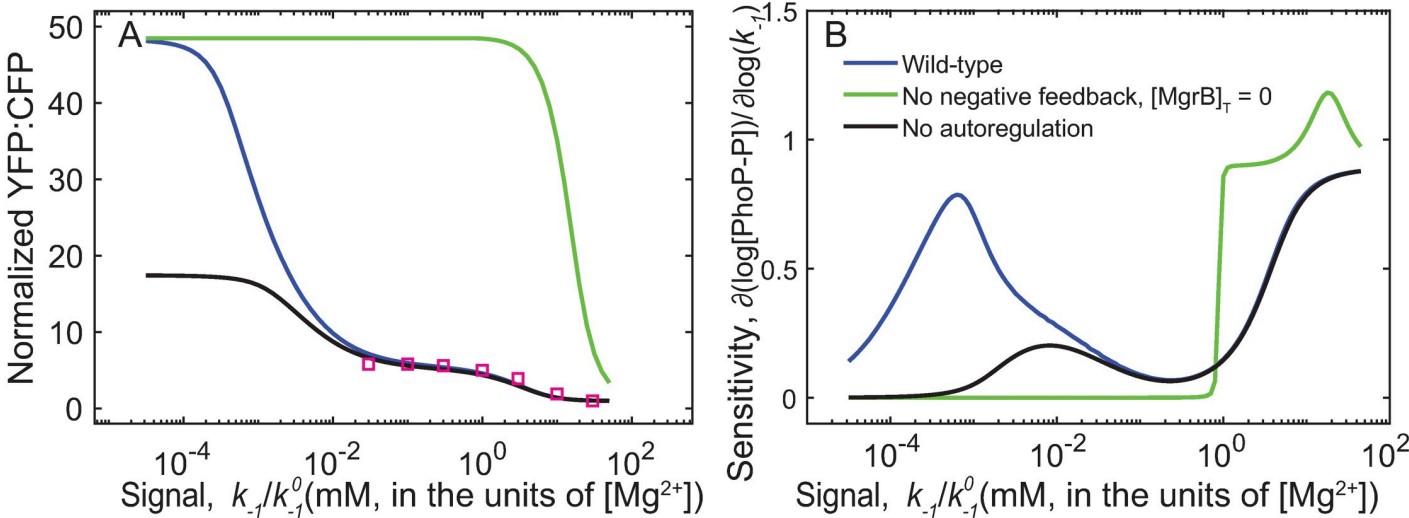

**Fig 5. Combination of positive and negative feedback increases range of sensitivity to signal.** A—steady state response of simulated promoter output (YFP, normalized to high $Mg^{2+}$ YFP) for models with no positive feedback (black), no negative feedback (green) and both feedback loops (blue). B- Absolute value of sensitivity (log derivative of PhoP-P with respect to $k_{-1}$, $\left| \frac{\partial(\log[\text{PhoP}-\text{P}])}{\partial(\log k_{-1})} \right|$) for model with no negative feedback (green) has the shortest range of signal sensitivity, while the wild-type model displays two phases of high sensitivity to signal.

## Combination of positive and negative feedback increases range of sensitivity to signal

What advantages does this unusual combination of positive and negative feedback provide? We know from experimental observations that negative feedback creates partial adaptation and faster kinetics, while positive feedback amplifies output and helps cells survive in growth limiting magnesium. To find out how steady state behavior is shaped by overlapping feedback loops, we simulated steady state dose-response with only one feedback present at a time.

We find a narrow range of signal sensitivity with negative feedback absent, while a much wider range without positive feedback (Fig 5A and 5B). Thus, in addition to kinetic advantages, negative feedback keeps the system sensitive to changes in magnesium over a much wider range of concentrations. However, without positive feedback the maximum output is much lower than with both feedback loops present, validating experimental observations of strong stimuli activating positive feedback. Taken together we find that negative feedback allows the TCS to tune the competing activities of PhoQ with time (to create overshoot dynamics) as well as stimulus (biphasic dose-response).

## Negative feedback can suppress oscillations

In addition to increasing range of sensitivity to signal, we unexpectedly find that negative feedback through upregulation of *mgrB* may also prevent oscillations in the network. When investigating the dynamics of the responses for *in silico* mutants lacking negative feedback, i.e. with constitutive *mgrB* expression(Fig 6A), we discovered limit cycle oscillations are observed at intermediate signal levels. These oscillations are only seen for low *mgrB* expression rates, i.e. with MgrB level comparable to that in unstressed wild-type cells (Fig 6B). Notably, these oscillations are observed for all the parameter sets that fit experimental data and show a biphasic dose response for wild-type cells (S3 Text). However, the oscillations were absent for the wild-type dynamics, i.e. when negative feedback is present(Fig 6B, blue line). The result is

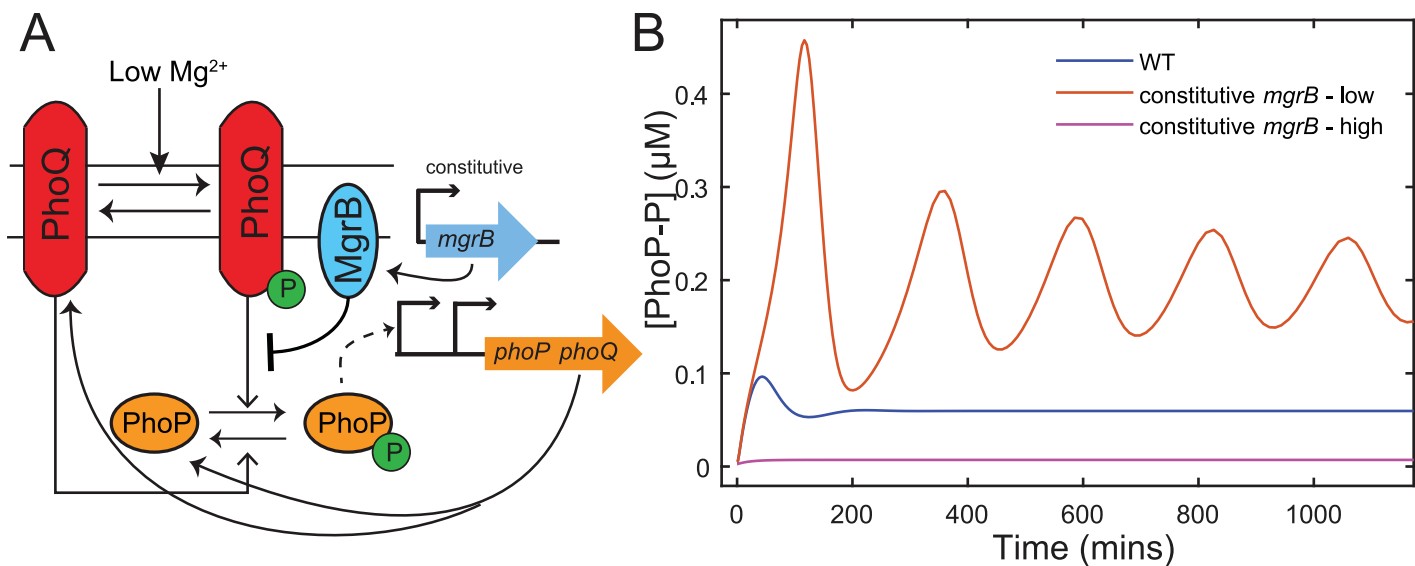

**Fig 6. Removing upregulation of mgrB can result in oscillations in the two component system.** A—Model schematic of an *in silico* mutant expressing *mgrB* constitutively instead of being expressed from the PhoP-P dependent promoter $P_{mgrB}$ B—Simulations of PhoP-P following a switch from high (50 mM) to intermediate (1 mM) Mg$^{2+}$ show oscillations if *mgrB* is expressed at constant but low levels. Oscillations are absent in wild-type models, as well as models expressing *mgrB* at a constant high rate.

unexpected since oscillations require negative feedback and in our case elimination of negative feedback leads to oscillations.

To understand the mechanism of the oscillations and why elimination of MgrB feedback create these we constructed a simplified model of the circuit (see S3 Text for detailed analysis). The result showed that oscillations arise from autoregulated expression of *phoPQ* operon, namely from PhoP-P dependent increase in PhoQ concentration. Indeed, since PhoQ is a bifunctional enzyme, autoregulation results in simultaneous positive and negative feedback [26, 27]. However, if PhoQ is first produced in a kinase conformation [16] and slowly switched to a phosphatase conformation, there will be a time-delay between the positive and negative component of the feedback. The fast positive and slow negative feedback leads to oscillations. Given that slow switching relative to effective MgrB-PhoQ binding rate ($k_b[\text{MgrB}]_T \gg k_{-1}$, S4 Text) at intermediate Mg$^{2+}$ is essential for our model to produce a plateau in dose-response, the time-delay and the resulting oscillations appear to be a robust prediction for constitutive *mgrB*. However, increase in MgrB concentration increases the fraction of PhoQ that is MgrB bound, speeding up PhoQ conversion to phosphatase state and thereby prevents oscillations by reducing the delay in the negative feedback. This is why oscillations are not observed when MgrB is upregulated by PhoP-P or when constitutive production of MgrB is too high (Fig 6B). Thus, we find that autoregulated PhoPQ TCS may use negative feedback through MgrB in order to avoid sustained oscillations in response to stimulus. Notably the oscillations are not a consequence of any particular model assumption, but rather seem to stem from a general mechanism that can be applicable to many autoregulated TCSs (as long as kinase conformation of the sensor is produced first and then slowly switches to phosphatase conformation). It remains to be seen if this mechanism can lead to oscillatory or pulsatile response for systems where it is physiologically beneficial.

## Discussion

The output of some two-component systems with bifunctional kinases—phosphorylated response regulator protein—displays robustness to overexpression of the two proteins. Here we

show that this property can extend to PhoPQ TCS which regulates the gene encoding MgrB that inhibits PhoQ kinase activity resulting in a negative feedback loop. Using models of PhoPQ TCS we show that PhoP-P can be insensitive to overexpression of PhoPQ if MgrB is expressed in excess of PhoQ. The PhoPQ TCS steady state response to decreasing $Mg^{2+}$ concentration displays two distinct phases of activation separated by a plateau. We propose roles for $Mg^{2+}$ and MgrB in modulating PhoQ activity such that a model recreates the biphasic nature of steady state dose-response. We propose that $Mg^{2+}$ binds to PhoQ and promotes the phosphatase conformation. Limitation of $Mg^{2+}$ then drives a change in conformation of PhoQ to the kinase form. We also hypothesize that MgrB suppresses the rate of this conformation change and the autokinase activity of PhoQ. Next, we find approximate analytical solutions for PhoP-P at different ranges of $Mg^{2+}$ concentration. In our models we find that strong MgrB-mediated suppression of rate constants and growth-dilution of proteins are important factors that shape the biphasic dose-response. Finally, we propose advantages gained by having such an overlaid feedback structure. Negative feedback limits activation at low but not growth limiting $Mg^{2+}$ and shapes a surge in transcription in response to large step downs in $Mg^{2+}$. Whereas positive feedback enables a strong activation of PhoP-P dependent promoters at growth-limiting $Mg^{2+}$.

With a third component modulating the kinase's activity, how does a PhoPQ TCS still obtain robustness to levels of the two components? Our analysis shows a possible condition in which PhoP-P can be robust to variations in PhoPQ expression. If MgrB is expressed at much higher levels compared to PhoQ, dependence of PhoP-P on total PhoPQ expression becomes negligible. This condition for robustness is not implausible within *E. coli* since $P_{mgrB}$ is one of the strongest PhoP-activated promoters [16] and is likely stronger than $P_{phoPQ}$. PhoQ, like many TCS sensor kinases, is expressed at low concentrations, estimated 50 fold less than PhoP [21, 25]. In fact, estimates of MgrB and PhoQ concentrations at $\sim 0.5$ mM $Mg^{2+}$ can be obtained from the database published by Li et al [25]. In rich medium, concentrations of MgrB and PhoQ are around 1.5 $\mu M$ and 0.2 $\mu M$ respectively. The near 7 fold difference is unlikely to decrease at lower $Mg^{2+}$ given estimated maximum fold activation of $P_{mgrB}$ is 60 [28], whereas fold activation of $\sim 20$ for $P_{phoPQ}$ can be computed from YFP measurements reported by Salazar et al [16]. It is possible that over the range of induction rates (upto 4x wild type) MgrB remains in excess of PhoQ.

Multiple PhoP-P dependent promoters plateau over a range of 1 to 0.01 mM $Mg^{2+}$ while remaining far from saturation [9], only to be stimulated strongly when $Mg^{2+}$ becomes growth limiting ($< 10^{-3}$ $mM$). How does PhoPQ output plateau at lower stimulus levels, but still maintain the ability to respond strongly when needed? Hypotheses of $Mg^{2+}$ binding PhoQ at multiple sites with differing affinities have been made [9], however our analysis uncovers a potential mechanism with fewer assumptions. Consistent with our assumptions, studies in *Salmonella* strains have suggested that $Mg^{2+}$ binding PhoQ increases its phosphatase activity. We propose a model in which $Mg^{2+}$ directly binds a kinase-conformation of PhoQ and switches it to phosphatase-conformation. Our analysis shows that if MgrB strongly suppresses autophosphorylation, as well as the switch from phosphatase to kinase conformation, dilution due to growth remains the only way through which PhoQ-MgrB phosphatase complex can decrease. This can create a regime where both phosphorylation and dephosphorylation of PhoP are independent of the signal rate (switching from kinase to phosphatase).

Finally, what are the advantages of encoding a negative feedback to limit activation of an autoregulated two-component system? PhoPQ two-component system is widely conserved across bacterial species including pathogenic bacteria such as *Yersinia pestis*, *Klebsiella pneumoniae* and *Salmonella typhimurium* [10, 14, 18–20]. PhoP-P regulons in these species also encode MgrB homologs that limit PhoQ activity [18]. Interestingly, mutation or otherwise inactivation of the *mgrB* gene was found to be the source of colistin resistance in *Klebsiella*

*pneumoniae* [19]. Conserved structure of the network suggests that the structure provides some fitness advantages in $Mg^{2+}$ limitation by controlling level of activation of PhoP-P regulon. In *E. coli* positive autoregulation of the PhoPQ TCS helps cells survive in growth-limiting $Mg^{2+}$. On the other hand, negative feedback creates a transcription surge in response to a step down in $Mg^{2+}$ concentration. Negative feedback also facilitates a faster response compared to a mutant strain expressing *mgrB* constitutively at levels such that steady state response of the two strains is comparable [16]. Other negative feedback designs can also provide some of the same benefits. Phosphate sensing PhoBR TCS in *E. coli* speeds response by encoding a negative feedback. In contrast with PhoPQ, PhoB-P does not upregulate a protein that suppresses kinase activity. Instead, PhoB-P represses the autoregulated *phoBR* promoter at high concentrations of PhoB-P [29]. This design can overcome the costs of positive autoregulation; however, this design does not create a transcription surge.

In *Salmonella*, benefits of a surge in transcription can be obtained independent of negative feedback through an MgrB-like protein post-translationally suppressing kinase activity [11, 26]. Models of the *Salmonella* PhoPQ TCS reveal how positive autoregulation and the phosphatase activity of PhoQ together create an initial surge and a later decrease in expression of genes in PhoP-P regulon. Interestingly, models are consistent with the observed loss of surge in transcription if *phoPQ* is expressed constitutively. In contrast, transcription surge is present with either constitutive or autoregulated *phoPQ* expression in *E. coli* [16]. The surge and subsequent decrease in transcription is lost in strains lacking *mgrB*, suggesting different mechanisms drive transcription surges in *E. coli* and *Salmonella*.

Post-translational negative feedback also helps maintain sensitivity of PhoQ over a wider range of stimulus levels. The *E. coli* PhoBR TCS does not show adaptation in response to a downshift in phosphate. The advantage of overlaid autorepression and positive autoregulation in PhoBR TCS is that it allows for selecting a stronger autoregulated promoter without sacrificing speed [29, 30], but limits maximum activation. For two TCSs that sense low level of two nutrients, what selection pressure could have led to evolution of these structurally similar but functionally different negative feedback loops remains unknown. Taken together, these findings show how interplay of positive and negative feedback can shape dynamical properties of the PhoPQ two-component system.

Using our models of PhoPQ TCS that can explain experimental observations, we can make some testable predictions. First, if the experiment measuring TCS output as a function of independent induction of *phoPQ* operon (Fig 1B) is conducted using a strain expressing *mgrB* constitutively, we predict that robustness should not be observed (S8 Fig). This lack of robustness of PhoP-P output to PhoPQ overexpression could become more apparent if *mgrB* is expressed at low levels. Second, with the same strain expressing *mgrB* constitutively, we find that oscillations are possible in the output when cells are switched from high to intermediate $Mg^{2+}$, eg. 50 → 1mM (S3 Text). Oscillations are not predicted if the cells are switched to low $Mg^{2+}$, i.e. 0.01mM or if *mgrB* is expressed at high levels (comparable to expression levels of wild-type cells at 0.01 mM $Mg^{2+}$, S6 Fig). Moreover, we observe these oscillations with *in silico* mutants of wild-type models that show a biphasic dose-response. The predictions of oscillation are relatively robust and no oscillations are observed with models that fit temporal data well but fail to display biphasic dose-response. These predictions must be tested in the future.

Notably, the uncovered oscillation mechanism is quite generic. It requires transcriptionally autoregulated TCSs with sensory kinase in two conformations, one kinase-dominant another phosphatase dominant. If both kinase and phosphatase states of the sensor kinase (SK) are increased proportionally with the increase in total-SK, the steady state RR-P is independent of total-SK, as has been seen in multiple experimental and theoretical works [22–24]. In other words, positive feedback (increase in kinase form of SK) exactly balances out a negative

feedback (increase in phosphatase form). While the above argument may hold true about steady state RR-P, positive and negative feedback may have different timescales. If sensory kinase in produced in the kinase-dominant conformation and then slowly switches to phosphatase-dominant one, sustained or damped oscillations are possible. Most natural TCSs are autoregulated and often activated by a ligand that drives a conformational change in sensory kinase [31, 32]. Therefore this oscillatory or pulsatile response dynamics may be observed in the systems where it is of physiological benefit and could be used in synthetic biology applications.

## Methods

### Model and simulations

Two mathematical models were developed to examine the dynamical properties of PhoPQ TCS. The first model considers a single bifunctional form of the kinase PhoQ, whereas the second model considers two separate conformations (kinase and phosphatase). A set of ordinary differential equations (ODEs) describes the rate of change of all protein and mRNA species (S5 Text). The phosphorylation and dephosphorylation cycle reactions follow previous models by Goulian and collaborators [21]. Gene transcription regulation is modeled by phenomenological models of (Hill-function) dependence on [PhoP-P].

   All models used (S5 Text) were simulated to follow experimental protocol as closely as possible. For time course, signal parameter (depending on model) was set to 1mM to compute an initial steady state vector of all state variables using ode15s in MATLAB. Using this as initial condition, signal was set at a pre stress value (50mM or 2mM) and integrated for 3.5 hrs. Then the signal was set to a post-stress value (0.01mM, 2mM or 10mM) and integrated for 2 hrs, at the same time points as the data. For steady state data, signal was set to the respective value and integrated to steady state.

### Experimental data

Experimental data was obtained from refs [16, 21]. Time course of YFP:CFP read out from plates following a switch from 50mM to 0.01mM $Mg^{2+}$ published in ref [16] was used to fine tune temporal parameters. YFP was either expressed from the PhoP-P dependent *mgrB* promoter or *phoPQ* promoter. Time course data for the following strains was used as constraint on parameters: wild-type ($P_{mgrB}$, $P_{phoPQ}$), *mgrB* deletion ($P_{mgrB}$, $P_{phoPQ}$), autoregulation deletion, autoregulation+*mgrB* double deletion. In addition, wild-type time-course data collected from $P_{mgrB}$ promoter for cells switched from 50mM to 10mM, 50mM to 2 mM and 2mM to 0.01 mM was also used. The value of YFP:CFP at t = 0 for wild type cells switched from 50mM to 0.01mM with YFP expressed from *mgrB* promoter was used to normalize all time-course data. Steady state YFP:CFP data with YFP expressed from $P_{mgrB}$ for a range of $Mg^{2+}$ concentrations (30mM to 0.03mM) published in ref [21] was used to fine tune steady state parameters. This steady state data was normalized to YFP:CFP at 30mM. All experimental data used for fitting is shown in S3 Fig.

### Error calculation

   **Time course.**   If $(t_i, y_i)$ is the experimental normalized YFP:CFP data for a given strain and $(t_i, y_i)$ represents simulated normalized YFP:CFP, then the squared residual error for time course is calculated as

$$E_t = \Sigma_i (y_i - \hat{y}_i)^2$$

Similar residual errors for all strains are then added together to give a total time course residual error $E_t$.

**Steady state signal response.**    Steady state simulated YFP:CFP at each signal level is normalized to the simulated value at 30mM ($\hat{x}_j^s$) and compared against the experimental value ($x_j^s$) leading to the residual error

$$E_s = \Sigma_i (\hat{x}_i^s - x_i^s)^2$$

In addition, steady state YFP:CFP is simulated at very high signal (equivalent of $10^{-4}$mM). If this value is not $> 6 \times$ [YFP:CFP(0.03mM)], then an error penalty of 25 (comparable to the maximum squared residual at 0.03 mM) is added to $E_s$.

## Parameter fitting

Parameters were fit to minimize the above squared residual error ($E_s + E_t$) using particle swarm optimization in MATLAB. Each particle swarm optimization run resulted in one parameter set. The best fitting parameter sets were used for further analysis.

## Supporting information

**S1 Fig. Steady state simulations of one-state PhoQ model of PhoPQ TCS.** Steady state [PhoP-P] as a function of total PhoP, PhoQ at various $[MgrB]_T^0$ levels (all concentrations in units of ($\mu M$). As $[MgrB]_T^0$ increases, a range of total-PhoP,PhoQ appears where PhoP-P is robust (B) Each point in (A) is an intersection of transcription and interaction modules (Fig 2A, main text). Red curve shows solution to the interaction module. At fixed total PhoP,PhoQ, total-MgrB is increased and steady state [PhoP-P] is computed. Black curve represents the saturating dependence of MgrB-total on [PhoP-P]. The dotted line is the analytical solution obtained by solving Eqs 1 and 2 in main text.
(PDF)

**S2 Fig. Robustness in one-state PhoPQ model insensitive to modest parameter variations.** (A,B) Phenomenon of robustness to PhoPQ variation is observed in the one-state PhoPQ model over varying λ, $K_D$ values or over varying ratio of PhoP:PhoQ. (C) Variations in PhoP: PhoQ ratio also do not change the conclusions of biphasic steady-state response of the two-state model.
(PDF)

**S3 Fig. Experimental data used for parameter fitting.** YFP:CFP ratio (normalized to the ratio at 50mM) extracted from [16] and [21]. This normalized data set was used to fit temporal and steady state parameters for all models described in this paper. Data was extracted using image analysis in MATLAB (except bottom right panel, which was read out manually from ref [21]). All data except constitutive *mgrB* (top right panel) was used to fit models.
(PDF)

**S4 Fig. Best fit simulations of dynamical model 1.** Simulations from a representative parameter set showing best quantitative fit for the simple PhoPQ model. Simulation of an in-silico mutant expressing *mgrB* constitutively (top right, solid line) at 10x basal transcription rate of *mgrB* in wild-type. This simulation in addition to PhoQ phosphatase-lacking mutant was used to verify whether a parameter set was accurate.
(PDF)

**S5 Fig. Verifying accuracy of dynamical model 2 by simulating *in-silico* mutant of PhoQ lacking phosphatase activity.** With parameters that generate the fit in S6 Fig, we simulate the $Mg^{2+}$ step-down with an *in-silico* mutant of PhoQ lacking phosphatase activity single mutant (light green, solid line) or double mutant with *mgrB*-deletion (dark green, solid line). These simulations show a qualitative match with the experimental data (dashed lines).
(PDF)

**S6 Fig. Best fit simulations of dynamical model 2.** Simulations from a representative parameter set showing best quantitative fit for the two-state PhoPQ model.
(PDF)

**S7 Fig. Biphasic dose-response of two-state model is not sensitive to ratio of maximum output to plateau output.** Simulations from a representative parameter set showing best quantitative fit for the two-state PhoPQ model with lower maximum output relative to plateau level.
(PDF)

**S8 Fig. Response of two-state model to overexpression of PhoPQ.** (A) Blue line shows simulated steady state expression from the $P_{mgrB}$ promoter at 1mM $Mg^{2+}$ from an *in silico* mutant with $P_{phoPQ}$ promoter under inducible control (instead of autoregulated). PhoPQ TCS model also predicts that PhoP-P output will not be robust to overexpression of PhoP and PhoQ if *mgrB* is expressed constitutively instead of under PhoP-P control (red line). Promoter expression is shown as YFP:CFP normalized to steady state WT YFP:CFP ratio at the same $Mg^{2+}$. The x-axis shows total [PhoP] from the *in silico* mutant simulation normalized to WT total [PhoP]. The axes are recreated from Fig 4C in [21], and blue triangles represent data from the same figure. (B) Over a larger range of total-PhoP, the two-state model also shows a range of PhoP expression in which PhoP-P does not vary significantly similar to one-state model.
(PDF)

**S1 Text. Analyzing PhoPQ-MgrB reaction network for absolute concentration robustness.**
(PDF)

**S2 Text. Steady state PhoP-P for one-state PhoPQ-MgrB model.**
(PDF)

**S3 Text. Model predictions and analysis of oscillations.**
(PDF)

**S4 Text. A framework to examine steady state signal response for two-state model of PhoPQ-MgrB.**
(PDF)

**S5 Text. PhoPQ TCS models: Reactions, ODEs and Parameters.**
(PDF)

## Author Contributions

**Conceptualization:** Oleg A Igoshin.

**Formal analysis:** Satyajit D Rao.

**Funding acquisition:** Oleg A Igoshin.

**Investigation:** Satyajit D Rao.

**Methodology:** Satyajit D Rao.

**Supervision:** Oleg A Igoshin.

**Writing – original draft:** Satyajit D Rao, Oleg A Igoshin.

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
