## [Decision Letter · Decision Letter 0]

16 Aug 2020

Dear Prof. Igoshin,

Thank you very much for submitting your manuscript "Overlaid positive and negative feedback loops shape dynamical properties of PhoPQ two-component system" for consideration at PLOS Computational Biology.

As with all papers reviewed by the journal, your manuscript was reviewed by members of the editorial board and by several independent reviewers. In light of the reviews (below this email), we would like to invite the resubmission of a significantly-revised version that takes into account the reviewers' comments.

We cannot make any decision about publication until we have seen the revised manuscript and your response to the reviewers' comments. Your revised manuscript is also likely to be sent to reviewers for further evaluation.

Sincerely,

Attila Csikász-Nagy

Associate Editor

PLOS Computational Biology

Mark Alber

Deputy Editor

PLOS Computational Biology

Reviewer's Responses to Questions

**Comments to the Authors:**

Reviewer #1: This manuscript investigated how the negative feedback of MgrB regulation impacts the robustness of E. coli PhoPQ system, and explored potential mechanisms for MgrB feedback to produce the biphasic response. Two-component system is the prevalent bacterial signaling system that attracts extensive computational modeling. PhoPQ is one of the best studied two-component system in terms of robustness, response dynamics and feedback control. The results presented in this study would be of importance to the general understanding of the bacterial signaling mechanisms.

One of the major focus is to model the biphasic response. However, the main issue is that the extent of biphasic response has not been well characterized experimentally. I’m wondering how the modeling results depend on the actual quantitation of the biphasic response, particularly the highest activity relative to the intermediate plateau.

In Fig. 1C, the YFP:CFP output of the second phase is approximately five fold of the intermediate plateau. This appeared not consistent with the experimental data in the cited reference (Ref. 19). In Ref. 19 (Miyashiro et al 2008 PNAS), the intermediate plateaued output was ~0.6 shown in Fig.1C while the steady state output at growth-limiting Mg2+ was ~1 (Fig.3A in ref. 19), less than two fold of the plateau at 0.6 in Fig. 1C. Thus, for the biphasic data modeled in this study, there is uncertainty associated with both the x-scale (growth-limiting Mg2+ not well defined) and y-scale (fold difference to plateau). How will this impact the modeling?

The following are my other comments:

1. Fig. 2, what is the unit of PhoPT concentration? Cp+Ct? It is better to state the unit in the figure legends. Can the modeling results be put into context of the actual measured cellular concentrations of PhoP?

2. I understand that the value of PhoP:PhoQ ratio will likely have minimal effect on the modeling conclusion. But I’m curious why a ratio of 40 was used instead of the actual measured value (7-18, Yeo et al Mol. Cell 2012; ~19, Li et al Cell 2014).

3. The impact of MgrB feedback on robustness clearly depends on the inhibition factor λ and MgrB-PhoQ affinity. Is it possible to comment how the values were chosen for the two parameters and how would they affect robustness?

4. Fig. 3, please label the reactions (blue arrows) with the constant k-1, this may help readers to better follow the texts.

5. I think the two-state model recapitulates accurately the conformation switching mechanism of histidine kinases. Can this model be used to explore robustness?

6. Line 291-293, it was predicted that the time-delay between positive and negative feedbacks gives rise to oscillations. The negative feedback here is the conformation switching from a kinase to a phosphatase. It is hard to imagine that the timescale of conformational switching would be significantly slower than the kinase reaction and protein expression. Thus oscillation appears very unlikely for mechanism alone.

7. Line 332, PmgrB is a weaker promoter than PphoPQ? So, there is a possibility that MgrB becomes no longer in excess to PhoQ when PhoPQ is fully activated at growth-limiting Mg2+?

Reviewer #2: The paper provides, via a mathematical model based on ODEs, a mechanistic explanation for the dynamical behaviour observed in PhoPQ two-component system. The models describe the interactions of the TCS with a negative feedback mechanism that is also regulated by the TCS response regulator. The model mechanisms are justified mainly by references to the papers 16,17,18,19. The results elucidate the effect of negative feedback on the TCS activity, in particular, the biphasic steady state response and suppression of oscillations.

The authors make simplifying assumptions, however they give careful explanations for the results. A strength of the paper is that the models are calibrated by experimental data from the references 16 and 19.

My main concern about the paper is the lack of an analysis of the parameter space: the claims are rather on the structural aspects of the system although the model is parameter-rich with about 20-22 rate constants, listed in Tables 1 and 2 in S1, and some other constants for the boundary conditions... Some of the choices for these values are somewhat justified in terms of fitting to experimental data and other assumptions. However, it is not clear how much the results rely on the rate constants and how much they are due to structural properties of the system, e.g., inhibition by MgrB or the two states of PhoQ. A sensitivity analysis, for example, would help to address this issue.

My feeling is that such a model could be fit to produce a variety of dynamic behaviours, including the ones in the paper. In my opinion the authors need to address why the presented models are more adequate in comparison to any other, which might again fit to experimental data with the availability of a large parameter space.

There are punctuation typos in the abstract, as well as the following:

positive feedback includes -> positive feedback, includes

Noun articles are missing as well.

There are grammatical mistakes, e.g., affect networks -> affects the network's

The following statement in the introduction is not accurate: "In absence of environmental stimuli, sensor kinases sometimes have phosphatase activity, i.e. they can dephosphorylate the response regulator." These TCS respond to starvation, so it is a kind of a reverse mechanism. This might require highlighting.

There is something wrong with the sentence in the "author summary": "our results show the advantages of such a combination feedback loops and predict the effect of their perturbation on the steady state and dynamic responses."

Reviewer #3: I found this to be an elegant and very interesting modeling study. The work is motivated by two concrete questions concerning the E. coli PhoQ/PhoP system—the fate of robustness when negative feedback on the kinase occurs and also the existence of biphasic behavior in dose-response curved. The authors come up with clear conclusions regarding both through an elegant combination of analytical and numerical studies. They also make a number of other interesting observations that follow from their two-state model. As the authors note, much if not most of their analysis is likely to be important for many other two-component systems. It is also notable that many predictions from this work can be tested experimentally.

I have a few comments that the authors should address.

1) lines 38-39 "PhoP activates transcription of a small lipoprotein (SlyB in 38 Salmonella and Yersinia pestis, MgrB in E. coli) that limits kinase activity" This sentence suggests that MgrB and SlyB are the same protein with different names in different organisms, or that SlyB is only in Salmonella and Yersina and MgrB is only in E.coli. However, they are different proteins and both are found in all three organisms. (MgrB was shown to affect PhoQ-regulated gene expression in all three organisms in ref 17.) Also, SlyB is a lipoprotein whereas MgrB is an integral membrane protein. The sentence should therefore be reworded.

2) Along, the same lines as above, lines 370-371 and lines 355-356 seem to imply that Yersina and Salmonella do not have MgrB and instead have other proteins that limit PhoQ activity. These sentences should therefore be re-worded. Also, the authors may want to mention Klebsiella since MgrB inactivation in this bacterium has received a great deal of attention due to its role in colistin resistance.

3) lines 99-100 and 104-108. The claim about ACR confuses me. In the Supplement S1 text, the reactions labeled A describe a model that looks to be the same as the model in reference 20 for EnvZ-OmpR. But in reference 20, it is shown that the model only leads to approximate robustness. In References 21,22, on the other hand, ACR is demonstrated for EnvZ-OmpR in the limit in which the reverse reaction from EnvZ-P to EnvZ is neglected . Are the authors claiming that the full model, with the reverse reaction rate k_{-ap} not set to zero, has ACR, and therefore the claim of approximate robustness of the model in reference 20 is incorrect? If so, then this needs to be explained in detail. If not, then the discussion of ACR should be revised.

4) There is relatively little discussion about K_D, the dissociation constant of MgrB-PhoQ. For the analysis of robustness, K_D was 0.1 in units of (Cp+Ct) (S2 text, p.3). How important is the magnitude of K_D for the approximate robustness? It would be helpful to include a few sentences discussing this.

5) For the two-state model used to explain biphasic behavior, the association and dissociation rates k2b and k2d in Table 2, p. 9 of S1 Appendix, look like they give a MgrB-PhoQ dissociation constant that is about 4000-fold lower than the value of used for the robustness analysis (0.1 vs ~ 4*10^(-4) ). Is there still a reasonable range of approximate robustness in this case for this much lower K_D? Also, it looks like there is a typo in the table - I assume these constants should be k2b and k2d not k1b and k1d.

6) If, in the two state model, MgrB lowers the rate of the PhoQ -> PhoQ* transition, then wouldn't this mean that for the model used to analyze robustness, MgrB should increase the phosphatase rate in addition to lowering the autophosphorylation rate. It seems to me that in an in vitro experiment one would see significantly faster dephosphorylation if high levels of MgrB are present. As far as I know, it is still an open question whether MgrB affects the phosphatase rate, so this is not necessarily a problem, but the authors should at least comment on this.

**Have all data underlying the figures and results presented in the manuscript been provided?**

Reviewer #1: Yes

Reviewer #2: Yes

Reviewer #3: Yes

PLOS authors have the option to publish the peer review history of their article (what does this mean?). If published, this will include your full peer review and any attached files.

Reviewer #1: **Yes: **Rong Gao

Reviewer #2: No

Reviewer #3: No
---

## [Decision Letter · Decision Letter 1]

15 Nov 2020

Dear Prof. Igoshin,

We are pleased to inform you that your manuscript 'Overlaid positive and negative feedback loops shape dynamical properties of PhoPQ two-component system' has been provisionally accepted for publication in PLOS Computational Biology.

Best regards,

Attila Csikász-Nagy

Associate Editor

PLOS Computational Biology

Mark Alber

Deputy Editor

PLOS Computational Biology

Reviewer's Responses to Questions

**Comments to the Authors:**

Reviewer #1: The manuscript has been greatly improved by revisions. Additional modeling results have nicely addressed all issues raised toward the previous version.

One of my main concern toward the original draft is that biphasic response is not well characterized quantitatively. Biphasic response has been described for phoPQ, but not explored in quantitative details. The two apparent activity plateaus shown in Fig. 3 of Ref. 21, which the authors based their estimation of biphasic responses upon, corresponded to the dynamic behaviors in response to a single growth-limiting Mg condition, not two steady states to different stimuli intensities. The revised manuscript demonstrated that the modeling conclusion is independent of the actual fold difference of the two steady states, and explored how other parameter values or structural features of regulation affect the robustness and biphasic nature of response. I greatly enjoyed reading the revised manuscript and do not have further comments.

Reviewer #2: The authors make a strong case of their claims by a careful analysis and they have responded to my and other referees' concerns with careful considerations. The mechanism they investigate explains two qualitative features of the steady-state dose-response curve in specific parameter regimes; those regimes that enable these features to arise. I agree that the predictions of the paper are worthwhile. I also agree with the authors that their model captures the phenotype only with a tight coupling of the model structure and its parameters. On the downside, the title as well as sentences like the one below suggest that the wiring of the network is the main source of the observed dynamics as in the cited ACR results.

"Taken together, this study  shows how mathematical modeling and experimental data can be used together to understand the relationship between network structure and cellular function in bacteria."

Reviewer #3: I am happy with the revised manuscript and the authors' response to reviewer comments.

**Have all data underlying the figures and results presented in the manuscript been provided?**

Reviewer #1: Yes

Reviewer #2: None

Reviewer #3: Yes

PLOS authors have the option to publish the peer review history of their article (what does this mean?). If published, this will include your full peer review and any attached files.

Reviewer #1: No

Reviewer #2: **Yes: **Ozan Kahramanogullari

Reviewer #3: No

---

## [Editor Report · Acceptance letter]

22 Dec 2020

PCOMPBIOL-D-20-01150R1 

Overlaid positive and negative feedback loops shape dynamical properties of PhoPQ two-component system

Dear Dr Igoshin,

I am pleased to inform you that your manuscript has been formally accepted for publication in PLOS Computational Biology. Your manuscript is now with our production department and you will be notified of the publication date in due course.

With kind regards,

Jutka Oroszlan
